# Using Carbonated BOF Slag Aggregates in Alkali-Activated Concretes

**DOI:** 10.3390/ma12081288

**Published:** 2019-04-19

**Authors:** Mohammad Mastali, Ahmad Alzaza, Khaled Mohammad Shaad, Paivo Kinnunen, Zahra Abdollahnejad, Bethany Woof, Mirja Illikainen

**Affiliations:** 1Fibre and Particle Engineering Research Unit, University of Oulu, Pentti Kaiteran katu 1, 90014 Oulu, Finland; Ahmad.Alzaza@student.oulu.fi (A.A.); khaledshaad1991@gmail.com (K.M.S.); Paivo.Kinnunen@oulu.fi (P.K.); Zahra.Abdollahnejad@oulu.fi (Z.A.); Bethany.Woof@student.oulu.fi (B.W.); mirja.illikainen@oulu.fi (M.I.); 2Department of Chemical Engineering, Loughborough University, Leicester LE11 3TZ, Leicestershire, UK

**Keywords:** mechanical properties, carbonation, freeze/thaw resistance, high temperature, drying shrinkage, efflorescence

## Abstract

This experimental study aimed to develop alkali-activated concretes containing carbonated basic oxygen furnace (BOF) slag aggregates. In the first stage, the impacts of replacing normal aggregates with carbonated BOF slag aggregates in different alkali-activated concretes were determined by assessing mechanical properties (compressive and flexural strengths), morphology, thermogravimetric analyses (TGA), differential thermogravimetry (DTG) and the crystalline phases using X-ray diffraction analysis. Second, the developed plain alkali-activated concrete was reinforced by different fibre types and dosages to limit the negative impacts of the drying shrinkage and to improve strength. Therefore, the effects of using different fibre contents (1% and 1.5% in Vol.) and types (Polyvinyl alcohol [PVA], Polypropylene [PP], basalt, cellulose and indented short-length steel) on hardened state properties were evaluated. These evaluations were expressed in terms of the compressive and flexural strengths, ultrasonic pulse velocity, mass changes, drying shrinkage and efflorescence. Then, the impacts of aggressive conditions on the hardened properties of fibre-reinforced alkali-activated concretes were evaluated under carbonation, high temperature and freeze/thaw tests. The results showed that using carbonated BOF slag aggregates led to obtain higher strength than using normal aggregates in alkali activated concretes. Moreover, the maximum enhancement due to reinforcing the mixtures was recorded in alkali-activated concretes with steel fibres.

## 1. Introduction

Due to increase of demand for construction materials, the building sector has increasingly been interested in finding more cost-effective and eco-efficient construction materials. Therefore, various studies investigated the feasibility of using industrial sidestreams in concrete production as replacement for Ordinary Portland cement (OPC). Alkali-activated binders showed high potential to replace cementitious binders for their lower carbon dioxide emissions, costs and mechanical and durability properties. Various aluminosilicate precursors could be used in these alternative binders, depending on the reactivities, costs and carbon dioxide (CO_2_) emissions such as fly ash, ladle slag, ground granulated blast-furnace slag and metakaolin [1,2,3]. In the alkali activation process, the aluminosilicate precursors are activated by alkalis, which often consist of a combination of sodium hydroxide (NaOH) and sodium silicate (Na_2_SiO_3_) [4]. 

Lately, efforts have been made to maximise the use of recycled materials in alkali-activated binders. One of these efforts is using different recycled aggregates in concrete. Steel slag aggregates have also been studied as a potential replacement material for natural and limestone aggregates, and concrete made with steel slag aggregate showed better compressive and flexural strengths, denser structure and better chemical resistance than natural or crushed limestone aggregate concretes [5,6,7]. Several studies [7,8,9,10,11,12] have reported similar or superior engineering properties of alkali-activated concretes over traditional cementitious-based binders in terms of mechanical properties, heat of hydration, quality and durability of the concrete, in addition to lower CO_2_ emissions and production costs. On the other hand, a reduction in the mechanical properties was reported when >25% of basic oxygen furnace (BOF) slag aggregate was used in the concrete mixture [7,13,14]. Moreover, some drawbacks such as high drying shrinkage and volumetric instability compared to OPC concrete have been observed in alkali-activated concretes [15,16,17,18]. The utilisation of BOF slag as aggregates is limited by its excessive free calcium oxide (f-CaO), which easily reacts with water or CO_2_, forming Ca(OH)_2_ or CaCO_3_ and causing severe expansion and volume instability [19]. A 5–10% volume expansion was reported in [20], which resulted in forming some cracks and a reduction in mechanical properties. Several BOF treatments were suggested and tested to limit the amount and effect of f-CaO. Pang et al. [21] and Bodor et al. [22] reported high efficiency of carbonation (temperature of 70–90 °C and 5–20 bar CO_2_ pressure for 2 h) in controlling f-CaO in BOF slag; in addition, 15–50% improvement in the compressive strength was obtained when carbonated BOF aggregates were used. Other treatment methods for controlling f-CaO include weathering [23], attrition and chelation [19], and water quenching [24]. Weathering is the least preferable treatment because of its long duration (≈ 6 months), and the calcite film forms a coat around the BOF slag aggregates that weakens the cohesion between the aggregate and matrix, resulting in a reduction of the mechanical and durability properties [7]. 

The use of BOF slag aggregates in alkali-activated concrete has been reported before. Palankar et al. [7,14] investigated the partial and full replacement of natural aggregate with carbonated BOF aggregates (by weathering for 6–9 months) in alkali-activated granulated blast furnace slag (GGBFS)/fly ash concrete. Additionally, the utilisation of carbonated BOF aggregates (through carbonation reactor) as a partial replacement of natural aggregates in cementitious mortars was studied in [22]. However, an extensive experimental study on using 100% carbonated BOF (by accelerated carbonation) as aggregates in fibre-reinforced alkali-activated GGBFS/ladle slag concrete has not been investigated yet. Therefore, to fill this knowledge gap, the current paper developed alkali-activated concretes incorporating carbonated BOF slag aggregates as well as a comparative examination of the effects of fibre introductions on the mechanical and durability properties of alkali-activated concretes. 

In the first stage of this paper, the effects of using carbonated BOF aggregates instead of natural aggregates will be investigated. In the second phase, the impacts of five different fibre types (polyvinyl alcohol, polypropylene, basalt, cellulose and steel) and corresponding dosages (1% and 1.5% in Vol.) on hardened state properties of the selected alkali-activated slag-based concretes will be studied. 

## 2. Experimental Plan

### 2.1. Materials and Mix Design

The mix compositions were comprised of ladle slag (LS), granulated blast furnace slag (GGBFS), basic oxygen furnace (BOF) slag, alkali activator and industrial fibres. The ladle slag and BOF slag were obtained from SSAB Europe Oy (Raahe, Finland), and Finnsementti (a Finnish company, Espoo, Finland) provided GGBFS. To obtain BOF aggregates, BOF slag was initially sieved (>500 µm) and then exposed to CO_2_ flow gas in the carbonation chamber (5% CO_2_ concentration, 60% RH, temperature of 23 °C) for 48 h. The calcium oxide content in BOF slag reacts with CO_2_ gas. This reaction leads to the formation of crystalline calcite [25]. Mastali et al. showed that carbonation improves the quality of recycled aggregates recovered from old OPC-based concretes [25,26,27,28]. Table 1 lists the chemical compositions of LS, carbonated BOF, and GGBFS measured by X-ray fluorescence (XRF, Oulu University, Oulu, Finland). Alkaline activator was prepared by mixing sodium silicate with molar ratio of 2.5 (Ms (SiO_2_/Na_2_O) = 2.5) and sodium hydroxide with molarities of 6M, 8M, and 10M. Sodium hydroxide was prepared by dissolving sodium hydroxide pellets in tap water 24 h prior to use and cooled at room temperature (23 ± 1 °C). In the present study, a constant ratio of 2.5 was considered for sodium silicate to sodium hydroxide ratio. Moreover, a constant alkali activator to binder ratios of 1 and 0.96 were chosen to activate the mix compositions in the first and second phases, respectively. The used normal sand in the first phase was “CEN standard sand” certified EN 196-1. The sand size distribution varied from 0.08 to 2.00 mm, and the maximum moisture content was 0.2%.

To show the efficiency of carbonated BOF aggregates, un-carbonated BOF slag aggregates were also initially used in different alkali activated concretes to compare and indicate the efficiency of using CO_2_ sequestration technique in the treatment of BOF slag aggregates. However, various cracks were formed on the specimens’ surface and formation of these cracks made the comparison between the effects of using un-carbonated and carbonated BOF slag aggregates very hard. Therefore, quartz-based aggregates were used to show the efficiency of CO_2_ sequestration technique in the treatment of BOF slag aggregates.

In the first phase, 36 different alkali activated concrete mix compositions were prepared as listed in Table 2. According to the experimental results, one alkali-activated binder containing carbonated BOF aggregate was chosen to be reinforced in the second phase of this work. Afterward, the selected mixture from the first stage was reinforced by 1% and 1.5% volume of each used fibre (polyvinyl alcohol [PVA], polypropylene [PP], basalt, cellulose and steel). Figure 1 depicts the fibres used to reinforce alkali-activated binders. Table 3 lists the mechanical and physical properties of the used fibres to reinforce the mix compositions. A plain and 10 reinforced mixtures were investigated. Their composition and designation were described in Table 4.

### 2.2. Casting and Curing

In the batching process, dry ingredients (LS, GGBFS and carbonated BOF [or normal sand in the first phase]) were mixed for 2 min. Afterward, alkaline activator was gradually added with an activator/binder (A/B) ratio of 0.96 (or 1 for the mixtures in the first stage (see Table 2)) and mixed for 3 min. Finally, the fibres were slowly introduced to the mixtures to avoid unfavourable impacts of adding fibres, such as agglomeration effects, and the mixture was stirred for 3 min. It is worth stating that fibres were not added to the mixtures proposed in the first phase of this paper.

For the first phase, 108 prismatic beams (40 mm × 40 mm × 160 mm) were cast and prepared to be assessed under mechanical testing. All specimens were sealed until the test day (28 d) and the sealed specimens were kept in the lab conditions (45 ± 5% RH, temperature of 23 ± 1 °C).

Moreover, eight specimens for each composition were cast into prismatic beams (40 mm × 40 mm × 160 mm), and two were prepared for measuring the drying shrinkage. The rest of the specimens were assigned to evaluate strength development under different conditions. The specimens were sealed with a plastic cover for 24 h until demoulding. Then, six demoulded specimens were resealed until the test day (28 d), while the assigned samples for assessing the drying shrinkage were kept at lab-controlled conditions without sealing (45 ± 5% RH, temperature of 23 ± 1 °C). It should be noted that the lab conditions were kept constant during the length measurements of the specimens.

### 2.3. Characterisation of Specimens

#### 2.3.1. Ultrasonic Pulse Velocity (UPV)

Ultrasonic pulse velocity (UPV) is an in-situ and non-destructive test used to assess the quality and coherence of the concrete [25,29,30]. This test evaluates the quality of concrete by measuring the velocity of an ultrasonic wave passing through the concrete between the two transducers (see Figure 2a). Higher velocities indicate more compactness of the concrete. The velocity of the ultrasonic waves is adversely affected by the porosity of the sample. Adding fibres to the concrete mixtures affects the compaction and porosity of the concrete. Therefore, to evaluate the effect of fibres on the compactness of the samples, the UPV test was used according to ASTM C597 recommendations [31]. The pulse velocity was measured by Equation (1):(1)$$V=\frac{L}{T}$$
where *V* is the pulse velocity (m/s), *L* is the distance between two transducers (m) and *T* is the transmission time (s).

#### 2.3.2. Flexural Strength

The flexural performance was assessed under Three Point Bending (TPB) loading (see Figure 3) according to the ASTM C78 recommendation [32]. Three prismatic beams were used to be tested under the flexural loading. In total, one hundred and eight prismatic beams (in the first stage) and 66 prismatic beams (40 mm × 40 mm × 160 mm) (in the second phase) were tested under flexural loading with a deflection rate of 0.6 mm/min to assess the effects of fibre reinforcement on flexural performance. Equation (2) was used to calculate the flexural strength:(2)$$\sigma_{f}=\;\frac{1.5\;FL}{bh^{2}}$$
where *F* is the flexural load (N), *L* is the span length (mm) and *b* and *h* are the width (40 mm) and height (40 mm) of the prismatic beams, respectively.

#### 2.3.3. Compressive Strength

According to the ASTM C349 recommendation [33], the two broken portions resulting from the flexural test can be used to assess the strength of mix compositions under compression loading. All samples were evaluated under a compressive load with a constant displacement rate of 1.8 mm/min using a testing machine with a load carrying capacity of 100 kN. The compressive strength of each mix composition was obtained by averaging the two portions. It is worth mentioning that three prismatic beams were tested under the flexural loading for each mix composition; therefore, the compressive strength of each mixture was obtained by averaging six broken portions resulting from the flexural test.

#### 2.3.4. Carbonation Test

To assess the impact of carbonation on fibre-reinforced alkali-activated compositions, specimens aged 28 days were placed in a carbonation chamber (5% CO_2_ gas concentration, 60% RH, temperature of 23 °C) for seven days prior to the testing days. Mass and UPV were also measured before and after exposing the CO_2_ gas to evaluate changes resulting from the carbonation. Finally, the carbonated beams were tested under both flexural and compressive loadings to investigate the carbonation effects on the strength development.

#### 2.3.5. High-Temperature Resistance

To investigate the effect of high temperature on the mechanical properties and quality of the plain and fibre-reinforced alkali-activated binders, two 28-days cured samples from each mix composition were heated in an oven with a temperature of 600 °C. The oven took three hours to reach the test temperature of 600 °C, and the temperature remained constant for 4 h. Similar to the carbonation test, the mass and UPV were measured before and after the test to assess changes resulting from heating the alkali-activated concrete. Then, the heated samples were submitted to the compressive and flexural tests.

#### 2.3.6. Freeze/Thaw Resistance

After 28 days of curing, the freeze/thaw test was conducted and specimens were vertically immersed in water so that 20 mm (half of the specimens) were in the water. The temperature varied from −20 °C to 15 °C; the test procedures were adopted based on defined protocol in [34]. The specimens were kept for 2 h at 15 °C and 2 h at −20 °C; changing temperature from 15 °C to −20 °C and vice versa took 2 h for each change, and this process comprised one cycle (see Figure 4). The test consisted of three cycles per 24 h. The mass and UPV of each specimen were measured before the test. After 60 cycles, the specimens’ masses and UPV were measured, and the specimens’ compressive and flexural strengths were also tested.

#### 2.3.7. Drying Shrinkage

According to the ASTM C157 recommendation [35], two prismatic specimens (40 mm × 40 mm × 160 mm) were prepared for each mixture to assess the rate of the drying shrinkage. Figure 2b depicts the used apparatus for measuring the length change. The first measurement was taken 24 h after mixing, while the rest of the measurements were taken daily during the first week and then twice a week until the stabilisation of length change was obtained.

#### 2.3.8. Efflorescence Rates

To observe the efflorescence potential of the mix compositions and the effect of the fibre types and contents, the 28 d-aged specimens were immersed in 20 mm of water (half of the specimens) for seven days in lab conditions (45 ± 5% RH and temperature of 23 ± 1 °C). Visual observations were monitored daily to note changes and ensure the level of water remained constant during the observation period.

#### 2.3.9. SEM Analysis

Scanning electron microscope (SEM, Oulu University, Oulu, Finland) was used to analyse the morphology of the samples. Chemical compositions and Micrographs were gathered at an accelerated voltage of 15 kV and 10 kV, individually, to keep the variable distance from 6 to 8 mm. Before SEM/EDS examination, all specimens were dried by using a low temperature (60 °C for 12 h) in an oven. Then, all specimens in the tiny cylinder were coated with a 30-nm-thick layer of platinum alloy.

#### 2.3.10. X-ray Diffraction (XRD) Analysis

An X-ray diffraction (XRD, Oulu University, Oulu, Finland) analysis was conducted to characterise the crystalline phases. First, the samples were crushed, sieved and their powders were used for this analysis. Every sample was scanned from 5° to 80° at a scan rate of 0.5 s per step. Phase identification was performed by employing the ICDD PDF4 database.

#### 2.3.11. Thermogravimetric Analysis (TGA) and Differential Thermogravimetry (DTG) Analysis

To study the reaction products and the conversion processes, a thermogravimetric analysis (TGA, Oulu University, Oulu, Finland) and differential thermogravimetry (DTG) were performed on the 28 d samples using the Precisa Gravimetrics AG. The samples were crushed, powdered and heated to 1000 °C at 10 °C/min in a nitrogen atmosphere to assess the thermal phase transitions under different temperatures.

## 3. Results and Discussion

### 3.1. Development of Alkali-Activated Slag-Based Concretes with Carbonated BOF Aggregates

The effects of using different contents of carbonated BOF and normal aggregates on the compressive and flexural strengths are shown in Figure 5. Generally, it was observed that using carbonated BOF as an aggregate significantly increased the strength compared to normal aggregate.

The results indicated that impacts of carbonated BOF aggregates depend on the precursor type. Maximum strength was measured in the mixtures incorporating both ladle slag and GGBFS, in which the maximum compressive strength was approximately 55 MPa in the mixture of G0.5L0.5-8M with a carbonated BOF-to-binder ratio of 3 (see Figure 5a). Additionally, it was revealed that increasing the ratio of carbonated BOF aggregates from 3 to 5 achieved a lower compressive strength. As shown in Figure 5a,b, the maximum influence of replacing normal aggregates with carbonated BOF aggregates on increasing the strength was detected in L1-6M with an aggregate-to-binder ratio of 5 (more than three times). The influences of different contents of carbonated BOF and normal aggregates on the flexural strength are illustrated in Figure 5c,d. Similar to the compressive strength, replacing normal aggregates with carbonated BOF aggregates increased the flexural strength, and the maximum flexural strength was recorded in the mixture of G0.5L0.5-8M (around 15 MPa).

The morphologies of the G0.5L0.5-8M mixture containing carbonated BOF and normal aggregates are shown in Figure 6a,b, respectively. Using carbonated BOF aggregates increased the calcium oxide content and provided a denser matrix than the mixture with normal aggregates. A higher CaO/SiO_2_ ratio justifies higher strength in the mixtures containing carbonated BOF aggregates than normal aggregates (quartz-based sand). Additionally, some cracks formed at the interfacial transition zone (ITZ) between normal aggregates and the matrix, which could degrade mechanical strength.

As discussed earlier, regardless of the sodium hydroxide molarity, it was found that a precursor consisting of a combination of ladle slag and GGBFS results in maximum strength. Therefore, a comparative XRD analysis was conducted in this phase to determine the impacts of forming different crystals in alkali-activated ladle slag-based binders, alkali-activated GGBFS-based binders, and alkali-activated ladle/GGBFS slag-based binders.

The mineral patterns of the mixtures containing carbonated BOF and normal aggregates and activated with sodium hydroxide of 8M are shown in Figure 7. The result indicates that similar XRD patterns are formed in the mixtures, regardless of differences in precursor type. However, the precursor type (ladle slag, GGBFS and ladle slag/GGBFS) affected the intensity characteristics of the diffraction peaks, and higher diffraction peaks formed in G0.5L0.5-8M. Comparing the mixtures with carbonated BOF and normal aggregates shows that carbonated BOF significantly increased crystallisation and changed the formed chemical products. These impacts could affect the morphologies of the mixtures, which this finding was in line with the SEM analysis.

The increased crystallinity could either enhance or degrade mechanical properties, depending on whether enough space exists in the mixture to be filled by the formed crystalline phases. The transition of amorphous gels to more ordered structures causes the local microstructure to change, introducing internal stresses [36]. The matrix would be denser if enough space is available for the formed crystalline, otherwise crystallisations form micro-cracks (as observed in Figure 6b). The differences in strength development may be justified by differences in the microscopic analysis. Furthermore, similar micro-cracks could be observed in the microstructure of the cement pastes with the addition of micro silica; these micro-cracks mainly come from increasing the crystallisation pressure of hydrated calcium silicates gel [37,38].

Figure 8 depicts the weight loss (due to increase of temperature up to 1000 °C) and derivative weight of the mixtures. Similar to other microscopic analyses, using carbonated BOF aggregates greatly affected the TGA and DTG curves. Weight losses in TGA could be considered in temperatures between 100 °C to 800 °C [39]. Both free water and structurally bonded water are available in the first stage. The free water could be evaporated up to 100 °C, and the weight loss from 100 °C to 800 °C is attributed to the structural water [39]. The mass loss rate slowed after 250 °C owing to the chemically bonded water and OH groups [40].

According to the TGA results in Figure 8a,b, higher mass loss was recorded in alkali-activated concretes containing carbonated BOF aggregates compared to normal aggregates. It shows that the mixture with carbonated BOF aggregates possesses more chemical products than the mixtures incorporating normal aggregates.

Regarding the DTA curves in Figure 8c,d, three major endothermic peaks at approximately 180 °C, 500 °C and 780 °C were observed in the mixtures with carbonated BOF aggregates; the large shoulder below 200 °C is contributed to the dehydration of the calcium-rich silicate (C-S-H) gel [41]. The second major endothermic peak could be assigned to Portlandite (Ca(OH)_2_) [42]. The third destruction phase (at 780 °C) could be attributed to the decomposition of calcite (CaCO_3_) [42]. The differences in the major endothermic peaks of the DTA curves indicate that additional gel formation may have occurred owing to carbonated BOF aggregates, since only one major endothermic peak at approximately 170 °C was observed in the mixtures with normal aggregates.

In the first phase of this paper, it was generally observed that using carbonated BOF aggregates had greater impact on increasing mechanical strengths of alkali-activated concretes than using normal aggregates. The G0.5L0.5-8M mixture with an aggregate-to-binder ratio of 3 demonstrated the maximum strength. Therefore, this mix composition was chosen to be reinforced for the next phase of this paper.

### 3.2. Comparative Effects of Different Fibres on the Hardened State Properties

After developing the plain alkali-activated slag-based binders containing carbonated BOF aggregates in the first phase of this work, this mix composition was reinforced by using different fibre types and dosages. In this section, the effects of the reinforcement of mix compositions on hardened state properties were investigated and the results were interpreted.

#### 3.2.1. Ultrasonic Pulse Velocity (UPV)

Figure 9a,b show the effects of adding different fibres on the UPV and mass of the different mix compositions, respectively. Since UPV can assess the quality and compactness of the concrete, it can also evaluate the efficiency of the fibres on increasing the porosity of the concrete. According to the results, the fibre types and contents affect the UPV and mass. The addition of the fibres reduced the UPV and mass of all mix compositions significantly, except mixtures of PVA1, PP1, and basalt1. For instance, the maximum reductions of UPV and mass were approximately 20% and 5%, respectively, which were observed in the mixtures reinforced with 1.5% basalt fibre. The reduction in the UPV and mass can be attributed to the increased number of air voids because of the addition of fibres, and the same observation was noticed in [29].

According to the results, UPV reduction is directly proportional to mass reduction and the increase of fibre volume fraction for all fibres except steel fibre. In the reinforced mixtures with PVA fibre, increasing the fibre content from 1% to 1.5% decreased the UPV and mass by about 5% and 3%, respectively. In the PP1.5 mixture, reductions of approximately 10% and 3% were observed in UPV and mass, respectively. The UPV and mass of the basalt fibre mixtures decreased by approximately 15% and 5%, respectively, with increasing the fibre content. Moreover, limited changes were noticed in the reinforced mixtures with cellulose fibre due to the increase in fibre content (from 1% to 1.5%), whereas a 3% and 2% reduction of UPV and mass were detected, respectively. The UPV loss of the mixtures reinforced with steel fibre was unaffected by increasing the fibre content from 1% to 1.5%, while both St1 and St1.5 mixtures showed approximately 15% UPV reductions. However, increasing the steel fibre content resulted in a 10% mass gain.

#### 3.2.2. Compressive Strength

The impacts of fibre inclusion on the compressive strength of the mixtures are shown in Figure 10. The addition of fibres to the mixtures did not specifically increase or decrease the compressive strength. Fibres influence the compressive strength negatively by increasing the air voids in the mixtures or positively by limiting the crack propagations [30]. According to Figure 10, the fluctuation in the effects of fibres is highly dependent on the fibre type and content in each mixture. Using different fibres with different physical and mechanical properties affected the bond properties at the fibre/matrix interface as well as the mechanical anchorage. According to the results, the maximum increase of the compressive strength was measured at around 60% (65 MPa) in both the PVA1 and St1.5 mixtures. These enhancements come from the strong bond at the fibre/matrix interface, which limits the crack propagation under the compression load. Mastali et al. [29] investigated the effect of reinforcing self-consolidating concrete with recycled steel, industrial steel, PP and their combination on the physical and hardened state properties. Their findings also demonstrated that the mixtures reinforced with 1.5% industrial steel fibre have the highest compressive strength.

The maximum reduction in compressive strength of about 70% (10 MPa) was obtained in the basalt1.5 mixture. This reduction, which was the maximum strength loss, was mainly due to the provided high porosity in the basalt1.5 mixture (proven by the highest UPV reduction [20%]).

In the reinforced mixtures with PVA fibre, increasing the fibre content from 1% to 1.5% reduced the compressive strength by 30%. Similarly, reductions in compressive strength due to the increased fibre volume fraction were measured approximately 15% and 70% in the reinforced mixtures with PP and basalt fibres, respectively. The PP1 mixture indicated an approximate 35% increase in the compressive strength. A limited reduction (2 MPa) was measured due to an increased cellulose fibre volume fraction from 1% to 1.5%. However, both mixtures (Cel1 and Cel1.5) showed approximately 20% improvement in the compressive strength compared with the plain mixture. The reduction in the compressive strength, caused to the increase of fibre content, was mainly because of the higher porosity of the reinforced mixtures with 1.5% fibre content (which was proven by the corresponding lower UPV). On the contrary, increasing the fibre content of the steel fibre increased the compressive strength. Aydin et al. [16] studied the effects of using steel fibre on the compressive strength of alkali-activated slag/silica fume mortar. Their results revealed that increasing steel fibre content in the mixtures increased the compressive strength.

#### 3.2.3. Flexural Strength

Figure 11 presents the impacts of reinforcing the prismatic beams on the flexural strengths. According to the results, the flexural strength of the mixtures was governed by the fibre type and content. Similar to compressive strength, the maximum increase of the flexural strength was measured approximately 60% (20 MPa) in the reinforced mixture with 1.5% steel fibre, compared to the reference mixture (13 MPa). The flexural strength superiority of the 1.5% steel fibre-reinforced mixture was also proven in [29]. The enhancement of the flexural strength was attributed to the mechanical anchorage of the fibres. The maximum flexural strength reduction of approximately 60% (6 MPa) was achieved in the basalt1.5 mixture, which can be explained by the highest UPV reduction with respect to the plain mixture. Moreover, unfavourable impacts of fibre balling in mixtures, which affects the fibre bridging action, explains the reduction in flexural strength of the basalt1.5 mixture. The flexural strength of the reinforced mixtures with PVA, cellulose and steel fibres increased, regardless of fibre content. An increase of approximately 10–40% was achieved in the reinforced mixture with PVA. Additionally, the reinforced mixtures with cellulose and steel fibres indicated 6–25% and 35–55% improvements in the flexural strength, respectively. The strong chemical bond at the fibre/matrix interface explains the flexural strength enhancement in the reinforced mixtures with PVA, cellulose and steel fibres. Moreover, the increase in flexural strength of the reinforced mixtures with steel fibre is mainly due to the mechanical bond between the fibre and matrix and higher mechanical anchorage provided by the geometry of the indented short-length steel fibre. Mastali et al. [29] showed a higher flexural strength for the reinforced mix compositions using hooked-end industrial steel fibre.

On the contrary, the flexural strength decreased by 5% and 25–60% with the addition of PP and basalt fibres in the mixtures, respectively. The reduction of the flexural strength in the reinforced mixtures with PP fibres was mainly due to the smooth surface of the PP fibre, which led to a weak bond at the fibre/matrix interfacial zone during debonding [29].

Interestingly, a reduction was monitored in the reinforced mixtures with PVA and cellulose fibres when the fibre content increased from 1% to 1.5%, while the mixtures reinforced by PP fibre were unaffected by increased fibre content. Moreover, in line with the findings in [16], the flexural strength of the steel reinforced mixtures increased (5%) with increased fibre content.

#### 3.2.4. Drying Shrinkage

Figure 12 depicts the drying shrinkage of mix compositions. According to the results, the length stabilisation was measured for all mixtures after ≈ 850 h. In general, all mixtures reached 90% of their ultimate drying shrinkage after 250 h (almost 10 days) (see Figure 12a). The length changes varied from 3–10%, regardless of fibre type and content (see Figure 12b). The reference mixture showed a drying shrinkage approximately 7% compared to the initial length. The minimum rate of the drying shrinkage was obtained in the St1 mixture (≈ 3%), while the maximum drying shrinking was measured in the PVA1 mixture (≈ 10%), in respect to their initial lengths. Generally, introducing different fibres had no specific trend of decreasing or increasing the drying shrinkage in alkali-activated slag-based concretes. For instance, addition of PVA fibre increased the drying shrinkage by 9–37% compared to the reference mixture. Since a 10–60% reduction in the drying shrinkage was recorded compared to the reference mixture when PP, basalt, cellulose and steel fibres were used.

In addition, no consistent trend was found between the drying shrinkage and fibre dosage. For instance, using 1% PP fibre reduced the drying shrinkage about 50%, whereas 1.5% PP fibre increased the drying shrinkage ≈ 3%. However, an increase of 35% in the drying shrinkage was reported in the PVA1 mixture, while the PVA1.5 mixture increased the drying shrinkage by approximately 10%. On the other hand, the drying shrinkages for basalt and cellulose-reinforced mixtures were unaffected by increasing the fibre content.

#### 3.2.5. The Efflorescence Rates

Figure 13 illustrates the visual observations of the efflorescence rates of each mixture up to seven days. As the fibre type and content are the only differences between the mixtures, it was assumed that the variations of efflorescence rates could be attributed to the fibre type and content. According to the observations in Figure 13, fibres could have different impacts on the efflorescence rates, depending on the fibre type, length and shape. Fibres could physically provide a confinement around the specimens and affect the efflorescence intensity. The efficiency of this confinement could be governed by the fibre orientations and dispersions in the mix compositions. Therefore, different influences were observed on the efflorescence rates due to use of different fibres.

Regarding the efflorescence rates, all mixtures showed similar rates compared to the reference mixture with the insignificant impact of fibre on the rate of efflorescence during the first day (24 h), except for the steel fibre-reinforced mixtures. After seven days observation, cellulose fibre had the highest efficiency in limiting efflorescence, regardless of fibre content. PVA1 and PP1.5 fibres showed a high efficiency in reducing the efflorescence. In addition, no consistent trend was noticed between efflorescence rate and fibre dosage. As shown in Figure 9, increasing fibre dosage increased the porosity of the mixture (lower UPV), and it was claimed in [43] that lower porosity was accompanied by the lower efflorescence, and the opposite observation was noticed in this study, as PP1 had lower porosity and higher efflorescence than PP1.5. On the other hand, PVA1.5 displayed higher porosity and efflorescence than PVA1. The mixture reinforced with 1% basalt fibre indicated similar efflorescence rate compared to the reference mixture after seven days, whereas basalt1.5 had higher efflorescence rate.

### 3.3. Effects of Aggressive Conditions on Hardened State Properties

#### 3.3.1. Carbonation Test

The effects of carbonation on the UPV and mass of the plain and reinforced alkali-activated concretes are shown in Figure 14a,b, respectively. Penetration of carbon dioxide gas into the mixture led to the consumption of the calcium source and formation of calcite crystals. This resulted in lower porosity for all mixtures; hence, slight increases in UPV and mass could be observed.

For all carbonated mixtures, the effect of carbonation on the mass gain was limited (≈ 1%). The carbonated plain mixture showed an increase of 0.2% in the UPV compared to the uncarbonated plain mixture. The reinforced mixtures with PVA fibres indicated an increase of approximately 5% for the UPV, regardless of the fibre volume fraction, when compared with their uncarbonated conditions. However, a combination of both carbonation impacts and increasing PP and basalt fibre contents in the reinforced mixtures increased the UPV from ≈ 0.1% for 1% fibre content to around 3% for reinforced mixtures with 1.5%. A slight increase (≈ 0.1%) in the UPV of the reinforced mixtures with cellulose and steel fibres was also observed when exposed to CO_2_ gas, regardless of fibre content. The increase of the UPV and mass gains in all mixtures were mainly due to the formation of CaCO_3_, which resulted in filling the gaps and reducing the porosity.

The produced calcium carbonate crystals in the mixtures due to the CO_2_ reaction with CaO and Ca(OH)_2_ in alkali-activated concrete positively affected the compressive strength by filling the gaps and reducing the porosity of the mixtures. Therefore, denser reinforced mixtures provided a stronger bond with fibres at the fibre/matrix interface, resulting in flexural strength enhancement. However, the formation of CaCO_3_ generates internal stresses in the reinforced mixtures, resulting in internal micro-cracks. Because of the internal micro-cracks in the reinforced mixture, a degradation in compressive strength can be observed, as flexural strength deterioration is affected by the effects of carbonation on the chemical bond between the fibre and its surrounding matrix.

In the carbonated plain mixture, the compressive and flexural strengths increased by approximately 20% compared to the uncarbonated condition (see Figure 15). Maximum increases in the compressive and flexural strengths reached 65% and 45%, respectively, were measured in the basalt1.5 mixture. The maximum reduction for both compressive and flexural strengths was registered around 30% due to carbonation in the PVA1.5 and PP1.5 mixtures.

The compressive and flexural strengths of the reinforced mixtures with basalt, cellulose and steel fibres increased, regardless of fibre content. The reinforced mixture with basalt fibre increased up to 45–65% and 15–45% for the compressive and flexural strengths, respectively. An approximate 25% increase was found in the compressive strength of the carbonated Cel1 mixture, while the compressive strength of the carbonated Cel1.5 mixture was unaffected. Moreover, the flexural strength of carbonated Cel1 and Cel1.5 mixtures increased by 10%. The carbonated St1 mixture showed 15% and 5% improvements for the compressive and flexural strengths, respectively. However, the mechanical strengths of the St1.5 mixture were unaffected by carbonation.

As indicated in Figure 15a, the increases in the compressive strength of the plain mixture and reinforced mixtures with basalt, cellulose, and steel fibres are mainly due to the lower porosity of their carbonated mixtures. According the results presented in Figure 15b, enhancement of the flexural strength in the reinforced mixtures with basalt, cellulose and steel fibres indicated positive influences of carbonation on their chemical bond with the matrix, which increases the load carrying capacity of specimens during debonding. Moreover, the flexural strength gain in the reinforced mixture with the indented steel fibre was due to the higher mechanical bond and interconnectivity between the fibre and surrounding matrix, resulting from the increased compactness of matric due to the formation of CaCO_3_ crystals.

However, the reinforced mixture with PVA fibre revealed an approximate 20% reduction for the compressive strength after exposing the carbonation conditions. Moreover, the PVA1 and PVA1.5 mixtures showed 15% and 20% reduction for the flexural strength, respectively. Similarly, the carbonated reinforced mixtures with PP fibre underwent a 5–10% reduction in compressive and flexural strengths. The compressive strength reduction in mixtures reinforced with PVA and PP was mainly due to the internal micro-cracks generated by internal stresses caused by the formed CaCO_3_ because of the limited efficiency of the used fibres in introducing enough air voids. A reduction in the flexural strength was recorded in reinforced mixtures with PVA and PP fibres, indicating negative effects of carbonation on the chemical bond properties at ITZ of the fibre/matrix.

#### 3.3.2. Freeze/Thaw Resistance

Table 5 presents the mass change of each mix composition. The mass loss resulted from freeze/thaw is notable and well-known in OPC concrete [44,45]; therefore, it is worthwhile to monitor changes in the mass of alkali-activated concrete. According to the results shown in Table 5, all mixtures underwent a mass loss. The mass loss was governed by the fibre types and contents. The plain mixture resulted in a ≈ 1% mass loss in the freeze/thaw test. The Cel1 mixture had the minimum mass loss of ≈ 0.9%, and the basalt1.5 mixture returned with the maximum mass loss of ≈ 2.6%. Increasing the fibre content increased the mass loss for all fibre types, where steel fibres showed the lowest mass change (≈ 0.01%) due to the increase in fibre content from 1% to 1.5%. For instance, the mass loss of the PVA1 mixture was ≈ 1%, while a mass loss of ≈ 1.75% was observed in the PVA1.5 mixture. Similarly, the PP1 mixture lost 1.1% of its mass in 60 freeze/thaw cycles, while PP1.5 lost 1.3%.

In line with mass loss, Figure 16 illustrates the UPV before and after experiencing 60 freeze/thaw cycles. All mixtures showed a reduction in the UPV after 60 freeze/thaw cycles.

Regarding the fibre-reinforced mixtures, the highest UPV reduction (≈ 40%) was measured in the basalt1.5 mixture, as well as the maximum mass loss (see Table 5). Reinforced mixtures with steel fibre showed the densest structure against freeze/thaw cycles and led to the lowest UPV reduction of ≈ 1% compared to their UPV before the test. St1 mixture (3297 m/s) showed lower UPV reduction than the St1.5 mixture (3244 m/s). Increasing the fibre content resulted in increasing the UPV reduction under different freeze/thaw cycles for all fibre types except PP and cellulose fibres, which can be explained by the higher porosity resulting from the high fibre content. For instance, a UPV reduction of ≈ 10% was measured in the mixture of PVA1, while a reduction of ≈ 25% was observed in the mixture of PVA1.5. On the contrary, 10% and 5% UPV reductions were achieved by fibre-reinforced mixtures with PP and cellulose fibres, respectively, regardless of fibre content.

Figure 17a,b depict the compressive and flexural strengths of the mix compositions before and after experiencing 60 freeze/thaw cycles, respectively. A reduction was noticed in the compressive and flexural strengths for all mixtures. The results showed that the compressive and flexural strengths of the plain mixture were reduced by 20% (30 MPa) and 30% (8MPa), respectively. The maximum reductions of the compressive strength (≈ 25%) and flexural strength (55%) were obtained for the mixture of basalt1. The highest resistance and the lowest strength reduction were obtained for St1.5, where an approximate 5% reduction was measured for both compressive and flexural strengths. The reductions in compressive strength can be explained by the internal micro-cracks resulting from the increase in volume of frozen water; similar observations were found in [46]. Yawei et al. [46] investigated the freeze/thaw resistance of alkali-activated blast furnace slag concrete (ASC) and reported that the internal structural damage and induced micro-cracks were caused mainly by the higher volume of frozen water.

According to the results indicated in Figure 17, increasing the fibre content enhanced the freeze-thaw resistance, except for cellulose fibre where the Cel1.5 mixture showed a higher compressive strength reduction (≈ 20%) than the Cel1 mixture (≈ 5%). Moreover, increasing the fibre content in the mixture PVA1.5 resulted in a higher flexural strength reduction (≈ 35%) than in PVA1 (≈ 20%). Furthermore, the compressive and flexural strengths decreased approximately 15% and 25% for the mixture of PP1, respectively, while PP1.5 underwent 5% and 20% reductions.

With respect to the visual observations of the mixtures after contacting with water and experiencing 60 freeze/thaw cycles, no swelling was observed. This demonstrates the efficiency of the accelerated CO_2_ treatment for minimising this problem in the BOF slag.

To date, no experimental data are available indicating the efficiency of using accelerated CO_2_ treatment for minimising the swelling of BOF aggregates in alkali-activated concretes. Bodor et al. [22] investigated the swelling of carbonated and uncarbonated BOF aggregates in cementitious-based mortars. They used carbonated BOF under pressure and elevated temperature in a stirred batch autoclave reactor (90 °C, 20 bar CO_2_ pressure, for 2 h). Their findings showed that all mixtures prepared with uncarbonated BOF aggregates displayed swelling in the range of 11.4–91.7 mm after immersing the mixtures in water at 20 °C for 24 h and boiling water for 3 h. They also recorded approximately 16 mm of swelling in mixtures containing carbonated BOF aggregates (0.5–1.6 mm). The swelling in mixtures incorporating carbonated BOF aggregates was mainly due to insufficient carbonation of the slag [22].

#### 3.3.3. High Temperature Resistance

The impacts of high temperature on the UPV and mass losses are presented in Figure 18a,b, respectively. Similar to other reported results, the UPV and mass losses due to high temperature were governed by fibre type and content. Exposure to high temperatures led to a mass loss of ≈ 10% and UPV loss of ≈ 80% in the reference mixture. Minimum UPV reductions of 55% and 70% were assigned for the specimens of St1.5 and St1, respectively (see Figure 18a). Moreover, as indicated in Figure 18b, these mixtures showed the minimum mass loss (≈ 10%) when compared to the unheated condition. The PVA1.5 mixture exhibited the maximum UPV reduction (≈ 90%) and the maximum mass loss of ≈ 20% compared to its unheated state.

Based on the results obtained, increasing the fibre content from 1% to 1.5% had a minor effect on the UPV and mass loss, except for steel fibre. The PP, basalt, and cellulose-reinforced mixtures had similar mass loss (≈ 15%), but their UPV reduced by 85%, 75%, and 80%, respectively, regardless of fibre content. Higher UPV losses were observed in the reinforced mixtures with nonmetallic fibres (e.g., PVA, PP, cellulose), which could be justified by the formed micro-channels and air voids when the nonmetallic fibres melted in high temperatures [47].

Losses in the compressive and flexural strengths are indicated in Figure 19a,b, respectively. Reduction in strength was observed for all mixtures. Like the UPV and mass reductions, the losses in the compressive and flexural strengths were functioned by the fibre type and content. The compressive and flexural strengths were reduced by 85% and 95% in the plain mixture, respectively. In line with UPV and mass losses, the maximum reductions of 85–95% were measured for both compressive and flexural strengths in the mixtures reinforced with PVA, PP, cellulose, and basalt fibres, regardless of the fibre volume fraction. The minimum compressive and flexural strength reduction of ≈ 75% was registered for steel-fibre reinforced mixtures with 1% and 1.5% fibre dosage.

The primary reason for the high losses may be attributed to the thermal mismatch between shrink paste and expanded aggregates, which caused volume changes resulting in debonding the fibres from the matrix (depending on the temperature and fibre type), increasing the porosity and propagation of the cracks [48]. Higher reductions in the compressive and flexural strengths of the mixtures reinforced with PVA, PP and cellulose fibres are mainly because of their low resistance to high temperatures, as reported in [48]. Yermak et al. [48] indicated that PP fibres melted at 170 °C, and the same behavior was expected in PVA and cellulose fibres, resulting in losing the fibre action in resisting the crack propagations and transferring the tensile stress over the fracture surfaces. Moreover, melting the fibres increased the mixture porosity, which diversely affected the strength.

As shown in Figure 20, the steel fibres did not melt, and steel fibres could transfer the tensile stress across the fracture surfaces. Lourenço et al. [47] proved the efficiency of steel fibre in increasing the post-cracking resistance of concrete in high temperatures.

In addition, the visual observations indicated that in fibre-reinforced specimens with basalt fibre, no signs of burned fibre were detected, demonstrating a better performance of these fibres compared to the other nonmetallic fibres when exposed to high temperatures.

## 4. Conclusions

This paper reports the experimental results extracted from developing alkali-activated concretes containing carbonated BOF slag aggregates. Afterward, the developed plain alkali-activated concretes containing carbonated BOF slag aggregates were reinforced with different fibre types and dosages to minimise the drying shrinkage and improve the mechanical strength. With respect to the results obtained in this study, the following remarks could be highlighted:Using carbonated BOF slag aggregates improved both compressive and flexural strengths up to three times. The maximum strength in alkali-activated concretes with carbonated BOF slag aggregates was obtained when a combination of ladle slag (50% of binder mass) and GGBFS (50% of binder mass) was used. Using carbonated BOF slag aggregates provided a denser matrix than normal aggregates.The highest enhancements in the mechanical properties were gained in the reinforced mixtures with 1.5% steel fibre (increase of 60% both compressive and flexural strengths).The formation of crystals due to exposure to CO_2_ gas reduced the porosity and improved the mechanical strengths of the plain mixture.The addition of fibre enhanced the freeze/thaw resistance. The reinforced mixtures with steel fibre, regardless of fibre content, showed the lowest strength reduction (5%), while the reinforced mixtures with basalt fibres (1%) showed the maximum strength reductions.Under aggressive conditions (high temperature, carbonation, and freeze/thaw tests), the minimum strength reduction was recorded in the reinforced mixtures with steel fibre.Drying shrinkage was governed by fibre type and content. The St1 mixture demonstrated the minimum drying shrinkage and showed a reduction of 60% compared to the plain mixture.The visual observations revealed that cellulose fibre had the highest efficiency in limiting efflorescence after seven days.

## Figures and Tables

**Figure 1 materials-12-01288-f001:**
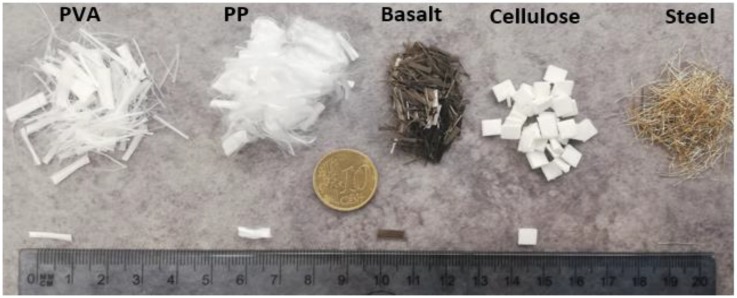
Fibres used for reinforcing the mixtures.

**Figure 2 materials-12-01288-f002:**
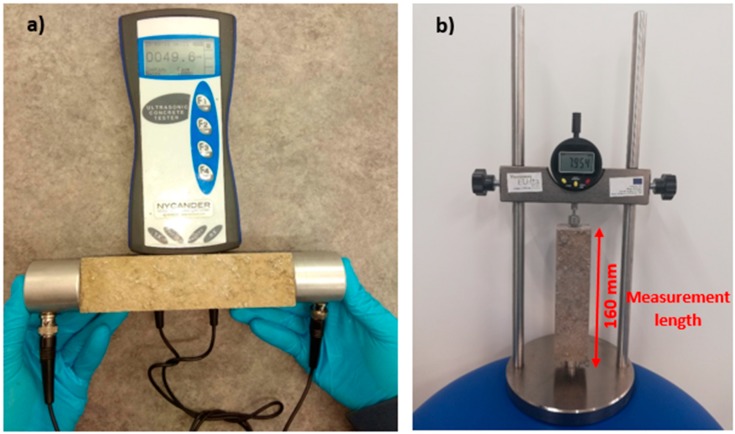
Devices used for measuring the: (**a**) Ultrasonic Pulse Velocity (UPV), (**b**) drying shrinkage.

**Figure 3 materials-12-01288-f003:**
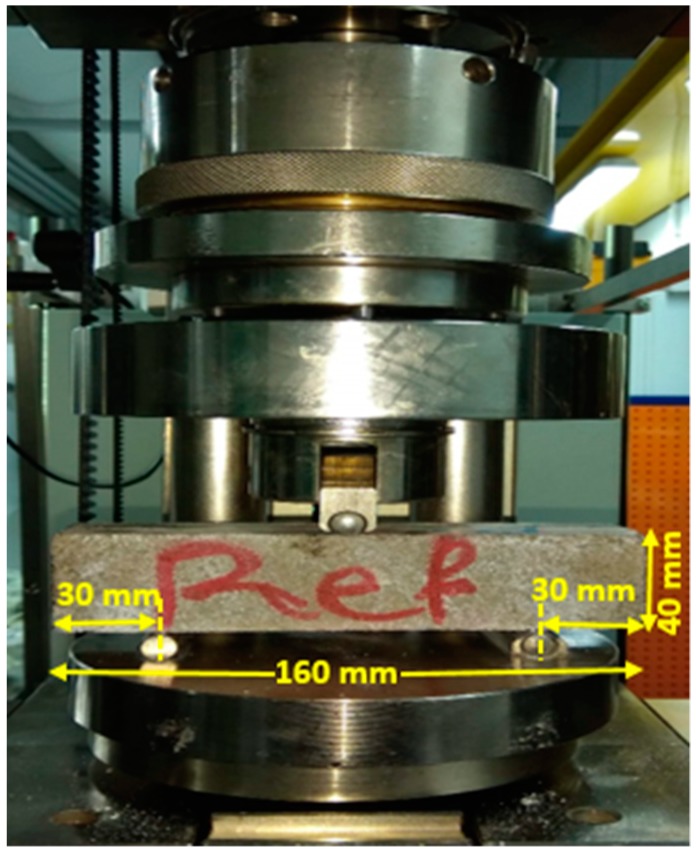
Adopted test setup for execution of the flexural test.

**Figure 4 materials-12-01288-f004:**
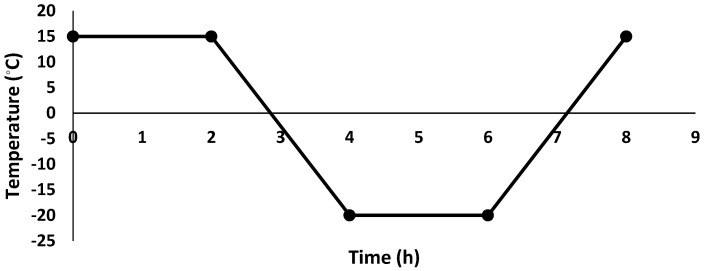
Freeze/thaw temperature changes per one cycle.

**Figure 5 materials-12-01288-f005:**
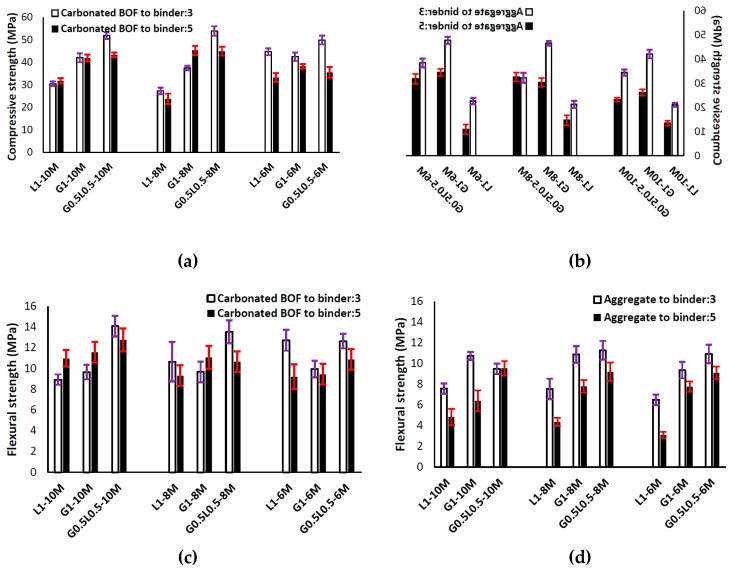
(**a**) Effects of using different contents of carbonated BOF on the compressive strength; (**b**) effects of using different contents of normal aggregate on the compressive strength; (**c**) effects of using different contents of carbonated BOF on the flexural strength; (**d**) effects of using different contents of normal aggregate on the flexural strength.

**Figure 6 materials-12-01288-f006:**
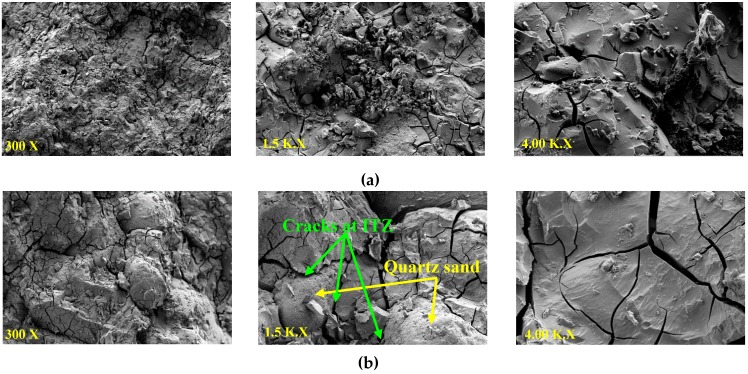
The morphology of the mixture G0.5L0.5-8M containing: (**a**) carbonated BOF aggregates; (**b**) normal aggregates (with aggregate-to-binder ratio of 3).

**Figure 7 materials-12-01288-f007:**
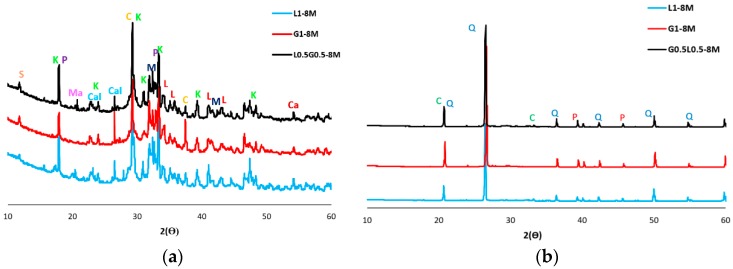
X-ray diffraction (XRD) patterns of mixtures with: (**a**) carbonated BOF (K: Katoite; P: Portlandite; C: Calcite; Ca: Calcio-olivine; M: Monetite; Ma: Mayenite; L: Larnite (C_2_S); Cal: Calcium hydrogen phosphate (Dicalcium phosphate); S: Srebrodolskite (Ca_2_Fe_2_O_5_)); (**b**) normal aggregate-to-binder ratio of 3 (C: Corundum; Q: Quartz; P: Paranatrolite).

**Figure 8 materials-12-01288-f008:**
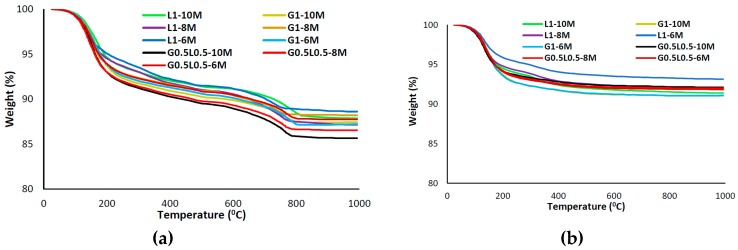
(**a**) Thermogravimetric analysis (TGA) in alkali-activated concretes containing carbonated BOF aggregate-to-binder ratio of 3; (**b**) TGA analysis in alkali-activated concretes containing normal aggregate-to-binder ratio of 3; (**c**) differential thermogravimetry (DTG) analysis in alkali-activated concretes containing carbonated BOF aggregate-to-binder ratio of 3; (**d**) DTG analysis in alkali-activated concretes containing normal aggregate-to-binder ratio of 3.

**Figure 9 materials-12-01288-f009:**
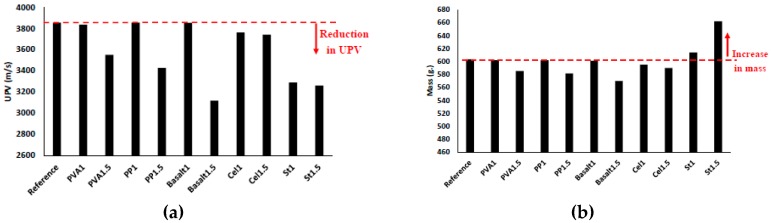
Effects of using different fibres and contents on the: (**a**) UPV; (**b**) mass.

**Figure 10 materials-12-01288-f010:**
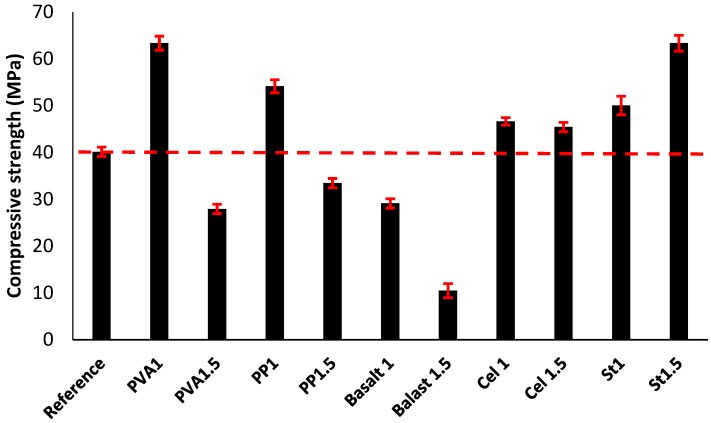
Effects of using different fibres and contents on the compressive strength.

**Figure 11 materials-12-01288-f011:**
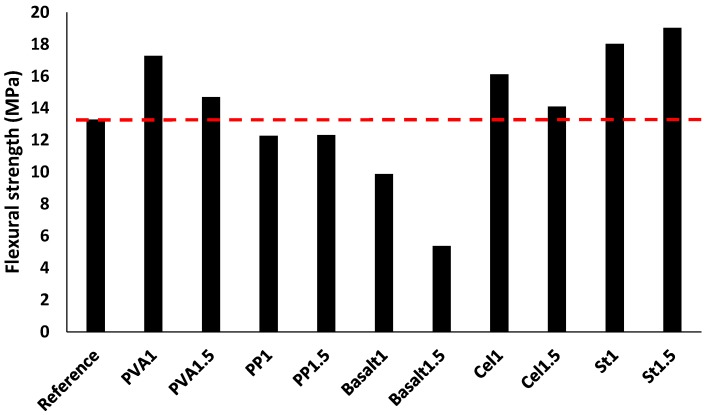
Effects of using different fibres and contents on the flexural strength.

**Figure 12 materials-12-01288-f012:**
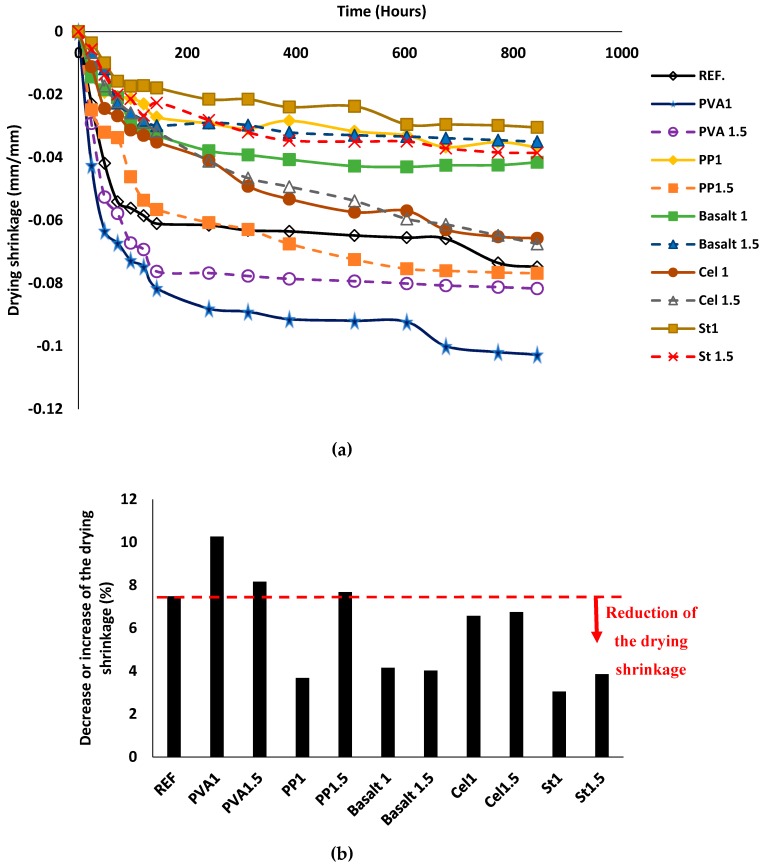
Impacts of using different fibres on: (**a**) the drying shrinkage of the mix compositions; (**b**) increasing or decreasing drying shrinkage.

**Figure 13 materials-12-01288-f013:**
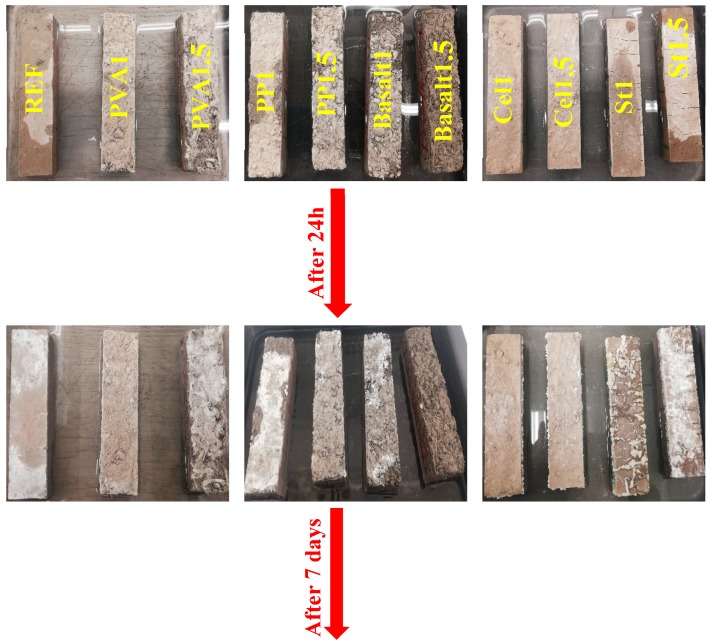
Visual observations from the impacts of using different fibres on the efflorescence rates.

**Figure 14 materials-12-01288-f014:**
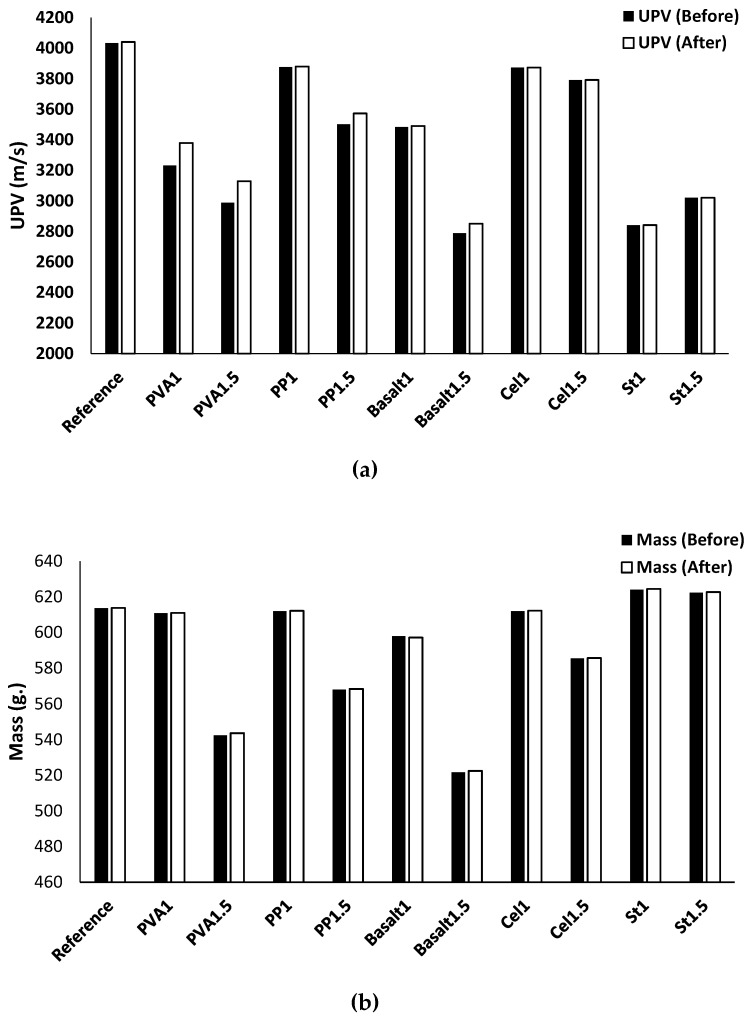
Effects of the carbonation on the: (**a**) UPV; (**b**) mass.

**Figure 15 materials-12-01288-f015:**
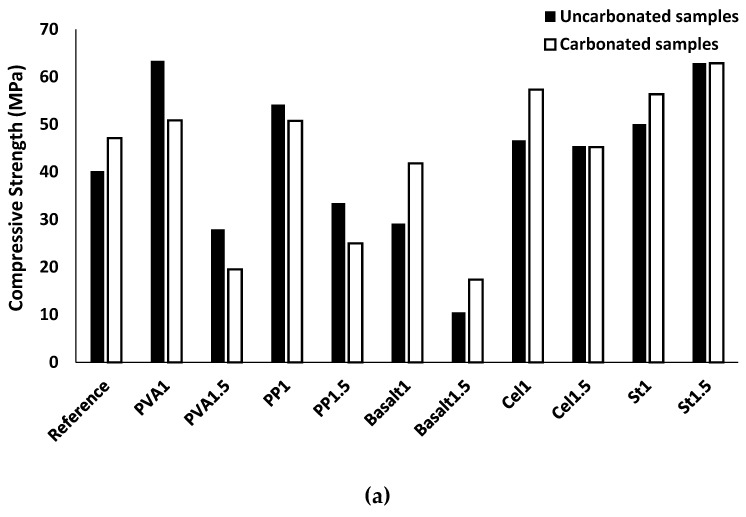
Effects of carbonation on the: (**a**) compressive strength; (**b**) flexural strength.

**Figure 16 materials-12-01288-f016:**
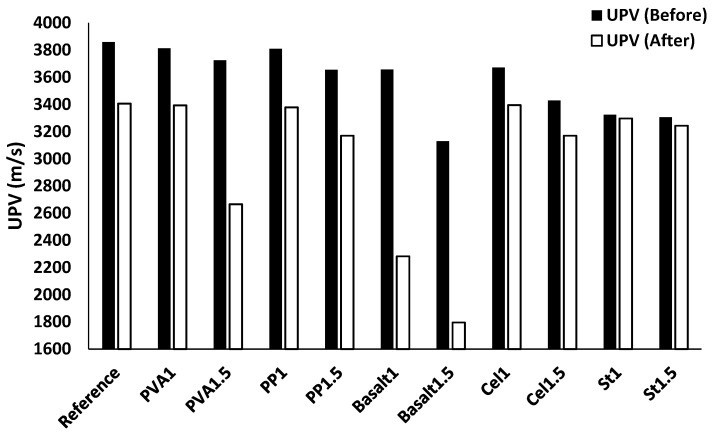
Effects of freeze/thaw conditions on the UPV.

**Figure 17 materials-12-01288-f017:**
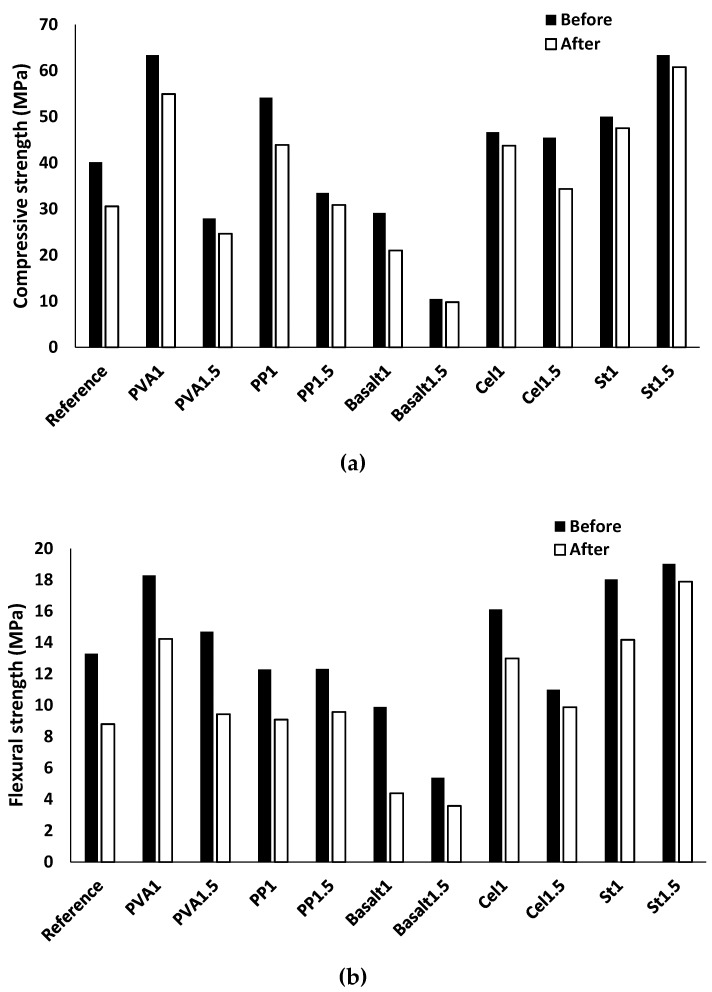
Effects of freeze/thaw conditions on the: (**a**) compressive strength; (**b**) flexural strength.

**Figure 18 materials-12-01288-f018:**
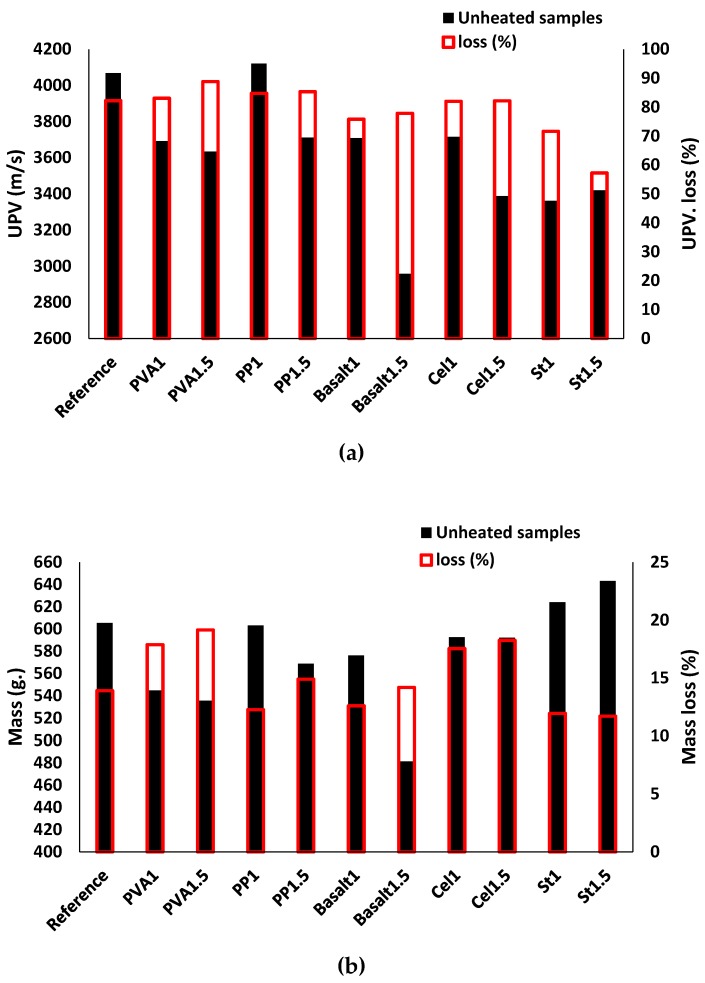
Effects of the high temperature (600 °C) test on the: (**a**) UPV; (**b**) mass.

**Figure 19 materials-12-01288-f019:**
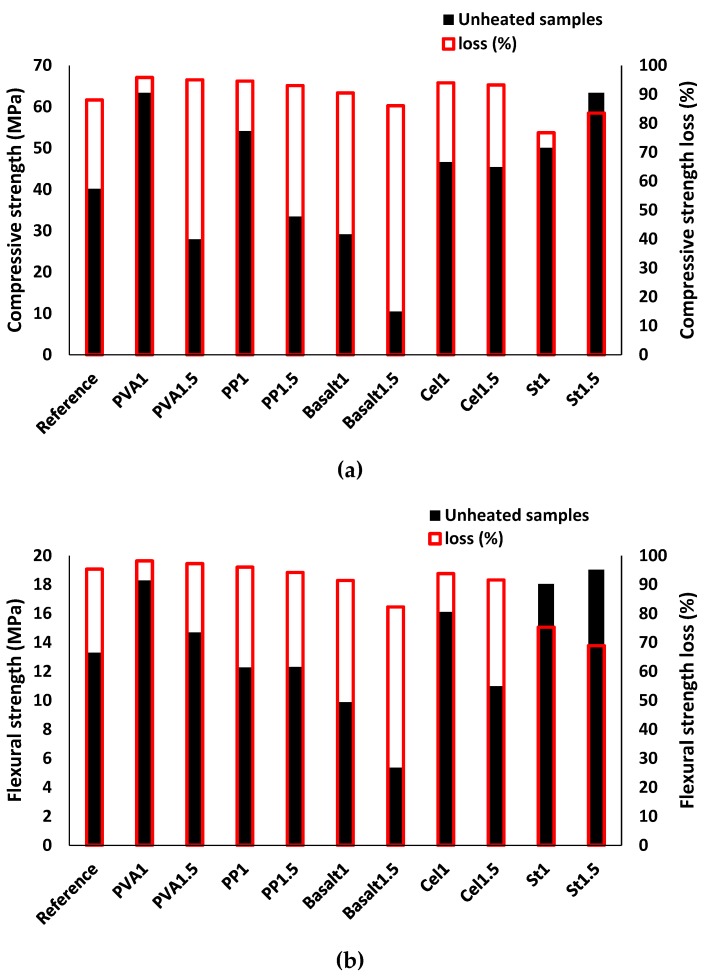
Effects of the high temperature (600 °C) test on: (**a**) the compressive strength; (**b**) the flexural strength.

**Figure 20 materials-12-01288-f020:**
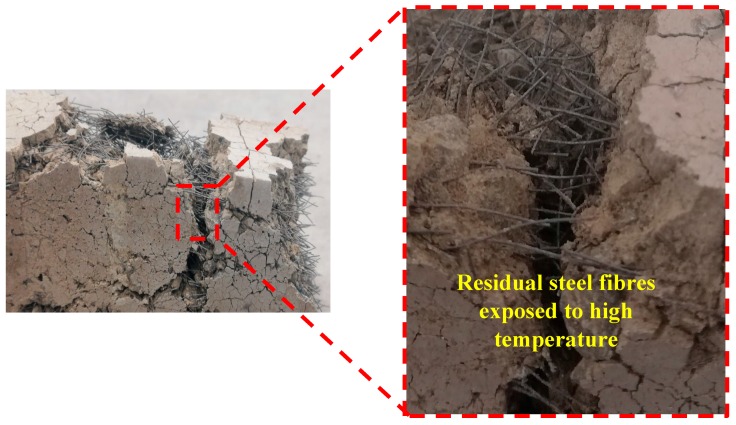
Residual steel fibres after experiencing high temperatures.

**Table 1 materials-12-01288-t001:** Chemical compositions of ladle slag (LS), carbonated basic oxygen furnace (BOF) slags, and granulated blast furnace slag (GGBFS) measured by X-ray fluorescence (XRF).

Material	Element/Oxides (%, w/w)					
	CaO	MgO	Al_2_O_3_	SiO_2_	SO_3_	Fe_2_O_3_
**LS**	50.96	6.31	27.87	8.27	0.8	1.13
**GGBFS**	38.51	10.24	9.58	32.33	4.00	1.23
**Carbonated BOF**	54.59	1.86	1.15	12.99	0.18	21.35

**Table 2 materials-12-01288-t002:** The proportions of the mixtures (% mass).

Specimen Designation	LD	GGBFS	Aggregates/B	A/B	SS/SH	SH (M)
**G1-6M**	0	1	3 or 5 (carbonated BOF slag or quartz sand)	1	2.5	6
**G1-8M**	0	1				8
**G1-10M**	0	1				10
**L1-6M**	1	0				6
**L1-8M**	1	0				8
**L1-10M**	1	0				10
**G0.5L0.5-6M**	0.5	0.5				6
**G0.5L0.5-8M**	0.5	0.5				8
**G0.5L0.5-10M**	0.5	0.5				10

**Table 3 materials-12-01288-t003:** Physical and mechanical properties of the fibres.

Fibre Type	Length/Diameter (mm/mm)	Elastic Modulus (GPa)	Tensile Strength (MPa)	Elongation at Break (%)	Density (g/cm^3^)
**PVA**	200	41.0	1600	6	1.30
**PP**	833	9.6	910	<12	0.91
**Basalt**	333	100.0	4500	3.1	2.63
**Cellulose**	117	8.5	750	-	1.10
**Indented steel**	47	200.0	1300	-	7.80

**Table 4 materials-12-01288-t004:** The composition of the alkali activated concretes (% mass).

Mixtures	$\frac{LS}{B}$	$\frac{GGBFS}{bB}$	$\frac{A}{bB}$	$\frac{SS}{SH}$	$\frac{BOF}{B}$	Fibre Type	Fibre Content (%, in Vol.)
**Reference**	0.5	0.5	0.96	2.5	3	---	0
**PVA1**						PVA	1
**PVA1.5**							1.5
**PP1**						PP	1
**PP1.5**							1.5
**Basalt1**						Basalt	1
**Basalt1.5**							1.5
**Cel.1**						Cellulose	1
**Cel.1.5**							1.5
**St.1**						Indented steel	1
**St.1.5**							1.5

**Table 5 materials-12-01288-t005:** Mass change (%) after experiencing 60 freeze/thaw cycles.

ID	Reference	PVA1	PVA1.5	PP1	PP1.5	Basalt 1	Basalt 1.5	Cel1	Cel1.5	St1	St1.5
**Mass change (%)**	−1.02	−1.02	−1.75	−1.12	−1.33	−1.71	−2.63	−0.93	−1.16	−1.52	−1.53

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
