# Peer review of "Using Carbonated BOF Slag Aggregates in Alkali-Activated Concretes"

_materials, 2019, doi:10.3390/ma12081288_

Reviewer 1 Report

The manuscript deals with an interesting topic. The work is appreciable for the significant amount of experimental data presented, however the manuscript is almost confusing and at time repetitive. A deep revision of the manuscript is necessary with reference to both its organization and the language.

All symbols, acronyms and labels used need to be defined at their first appearance in the text.

A more concise and to-the-point description of experimental procedures adopted is necessary. No comment/discussion of the results should be reported in section “materials and methods”. A clear presentation of figures and tables is necessary. Discussion of the results should follow their presentation.

More concise conclusions are recommended.

Some suggestions (non-hexaustive indications) are reported in the marked manuscript attached to this report.

Author Response

Reviewer 1:

Comment 1: The manuscript deals with an interesting topic. The work is appreciable for the significant amount of experimental data presented, however the manuscript is almost confusing and at time repetitive. A deep revision of the manuscript is necessary with reference to both its organization and the language.

Answer: First of all, the authors would like to thank the Reviewer for his/her thorough examination of the paper and valuable comments, which have contributed to improve the overall quality of the work. Next, all the comments and suggestions made by the Reviewer were considered and responded in the revised version of paper. Moreover, a native English speaker checked paper. It was also tried to remove the repetitive sections in the paper. 

Comment 2: All symbols, acronyms and labels used need to be defined at their first appearance in the text.

Answer: All symbols, acronyms, and labels used need to be defined at their first appearance in the text.

 Comment 3: A more concise and to-the-point description of experimental procedures adopted is necessary. No comment/discussion of the results should be reported in section “materials and methods”. A clear presentation of figures and tables is necessary. Discussion of the results should follow their presentation.

Answer: Since some explanations were presented before the experimental procedures, some sections in this part were removed from the paper. Moreover, the comments and discussions that were presented in the section “materials and methods” removed to their relevant sections. All presented symbols in the tables defined as well as the captions of tables changed based on the Reviewer’s suggestion.

The only point is about green background revealed in the version sent to the Reviewers. In the original version prepared by the authors, all figure’s backgrounds were white. Since the authors should submit the paper’s file as word files and the files’ conversion to pdf was executed by the journal, these types of mistake happened. In both revised and original versions of the paper, the figure’s backgrounds are white.

 Comment 4: More concise conclusions are recommended.

Answer: Conclusion made shorter and even some of conclusions, which were not supported by the results, were eliminated.

 Comment 5: Some suggestions (non-hexaustive indications) are reported in the marked manuscript attached to this report.

Answer: Most suggestions were considered in the paper. The changes were highlighted in the revised version. The Reviewer’s answers also replied in the attached pdf file. 

Reviewer 2 Report

please remove green background from your figures and also green lines -use markers in all figures -some of conclusions are not supported by the results -shorten your conclusion section and revise your abstract

Author Response

Reviewer 2:

Comments: Please remove green background from your figures and also green lines -use markers in all figures -some of conclusions are not supported by the results -shorten your conclusion section and revise your abstract.

 Answer: First of all, the authors would like to thank the Reviewer for his/her thorough examination of the paper and valuable comments, which have contributed to improve the overall quality of the work. Next, all the comments and suggestions made by the Reviewer were considered in the revised version of paper.

Green background revealed in the version sent to the Reviewers. In the original version prepared by the authors, all figure’s backgrounds were white. Since the authors should submit the paper’s file as word files and the files’ conversion to pdf was executed by the journal, these types of mistake happened. In both revised and original versions of the paper, the figure’s backgrounds are white.

Both conclusion and abstract were shortened. Moreover, the conclusions that were not supported by the results, were eliminated. 

Reviewer 3 Report

No suggestion to make!

Author Response

The authors would like to thank the Reviewer for his/her thorough examination of the paper.

Reviewer 4 Report

The paper concerns the investigations about the properties of alkali-activated concretes with BOF slag aggregates. The study is extensive, a lot of compositions have been tested, and a lot of tests have been performed. The structure of the paper is correct. The paper has a high practical as well as scientific value. There are some shortcomings, which should be corrected before acceptance the paper for publication. Detailed comments are listed below:

1.      The paper should be checked by native English speaker in order to improve the style; there is also few syntax and grammar errors.

2.      Please check the formatting. In some places it is not compatible with the journal requirements.

3.      Line 155-156 - the average of two results is definitely not enough to assess the reliability and reproducibility of the results.

4.      Line 159 - from how many samples the average tensile strength have been obtained? Please make a comment on this.

5.      All figures would be much more readable if it had a white background. In addition it would seem more professional.

6.      Section 2.3.9 - were samples before the SEM examination dried?

7.      Figure 5 and 10 - a basic measure of variability of results should be given, e.g., standard deviation or coefficient of variation. Without this data, the reliability of the results can not be assessed.

8.      Line 266-267 - a similar effect is sometimes obtained with a cement paste with the addition of microsilica. The lime absorption takes too fast in such case, which increases the crystallization pressure of hydrated calcium silicates and leads to micro-cracks in the material structure. Examples of references to the literature in this topic:

Heikal, M., El-Didamony, H., Sokkary, T.M., Ahmed, I.A. Behavior of composite cement pastes containing microsilica and fly ash at elevated temperature. Constr. Build. Mater. 2013, 38, 1180–1190.

Szeląg M.: Development of cracking patterns in modified cement matrix with microsilica. Materials, vol. 11(10), 2018, 1928

Author Response

Reviewer 2:

The paper concerns the investigations about the properties of alkali-activated concretes with BOF slag aggregates. The study is extensive, a lot of compositions have been tested, and a lot of tests have been performed. The structure of the paper is correct. The paper has a high practical as well as scientific value. There are some shortcomings, which should be corrected before acceptance the paper for publication. Detailed comments are listed below:

Comment 1:  The paper should be checked by native English speaker in order to improve the style; there is also few syntax and grammar errors.

Answer: The authors are agree with the Reviewer. A native English speaker checked paper.

 Comment 2:  Please check the formatting. In some places it is not compatible with the journal requirements.

Answer: The authors are agree with the Reviewer. In the first draft sent to the Reviewers, the authors did not consider the journal format. Since the revised version was prepared based on the format journal sent to the authors and the journal requirements were considered in the revised version. 

 Comment 3:  Line 155-156 - the average of two results is definitely not enough to assess the reliability and reproducibility of the results.

Answer: Thanks for your useful comment. However, the authors intended to show that two specimens could be just tested from one prismatic beam. However, three prismatic beams were tested under the flexural loading for each mix composition, therefore, the compressive strength of each mixture was obtained by averaging six broken portions resulting from the flexural test.

 Comment 4: Line 159 - from how many samples the average tensile strength have been obtained? Please make a comment on this.

Answer: Three prismatic beams were tested for obtaining the flexural strength of each mixture. A sentence was also added to the text to show how many specimens were tested in this evaluation.

 Comment 5: All figures would be much more readable if it had a white background. In addition, it would seem more professional.

Answer: Green background revealed in the version sent to the Reviewers. In the original version prepared by the authors, all figure’s backgrounds were white. Since the authors should submit the paper’s file as word files and the files’ conversion to pdf was executed by the journal, these types of mistake happened. In both revised and original versions of the paper, the figure’s backgrounds are white.

 Comment 6: Section 2.3.9 - were samples before the SEM examination dried?

Answer: Yes, they were heated at a low temperature (600C for 12 hours) in an oven. This sentence was added to the text.

 Comment 7: Figure 5 and 10 - a basic measure of variability of results should be given, e.g., standard deviation or coefficient of variation. Without this data, the reliability of the results can not be assessed.

Answer: The authors are agree with the Reviewer. Error bars were added to the mentioned figures.

Comment 8: Line 266-267 - a similar effect is sometimes obtained with a cement paste with the addition of microsilica. The lime absorption takes too fast in such case, which increases the crystallization pressure of hydrated calcium silicates and leads to micro-cracks in the material structure. Examples of references to the literature in this topic:

Heikal, M., El-Didamony, H., Sokkary, T.M., Ahmed, I.A. Behavior of composite cement pastes containing microsilica and fly ash at elevated temperature. Constr. Build. Mater. 2013, 38, 1180–1190.

Szeląg M.: Development of cracking patterns in modified cement matrix with microsilica. Materials, vol. 11(10), 2018, 1928

 Answer: Thanks for the useful comment. I add the mentioned point to show that these micro-cracks could be formed in the cementitious matrix. Moreover, the mentioned references were cited. 

Round  2

Reviewer 1 Report

Replace definition of C-S-H by “hydrated calcium silicate”

CaCO3 and Ca(OH)2 are not acronyms, I suggest you delete them from the list.

Footnote in table 2 is not necessary. Symbols are included in the acronym list.

Author Response

Comment 1: Replace definition of C-S-H by “hydrated calcium silicate”

Answer: First of all, the authors would like to thank the Reviewer for his/her thorough examination of the paper and valuable comments, which have contributed to improve the overall quality of the work. The authors are agree with the Reviewer. It was changed in the text, however, since the authors used this definition in Figure 8, once this term was defined in the text.

Moreover, a native English speaker checked paper again to remove of all typos and grammar errors.

 Comment 2: CaCO3 and Ca(OH)2 are not acronyms, I suggest you delete them from the list

Answer: They were removed.

 Comment 3: Footnote in table 2 is not necessary. Symbols are included in the acronym list.

Answer: They were removed.

Reviewer 2 Report

There are still a lot of typos and grammar errors in paper. Authors need to fix this problem. Please ensure all figures are explained deeply in the text.

Author Response

Comment 1:  There are still a lot of typos and grammar errors in paper. Authors need to fix this problem. Please ensure all figures are explained deeply in the text.

Answer: First of all, the authors would like to thank the Reviewer for his/her thorough examination of the paper and valuable comments, which have contributed to improve the overall quality of the work. The authors are agree with the Reviewer. A native English speaker checked paper again to remove of all typos and grammar errors.

Moreover, there were some figure’s numbers that did not exactly mentioned in the text (for instance, in the text, they did not assign which sections (a, b, c, or d) of a figure that we are talking about). Therefore, these numbers were exactly mentioned in the text and the readers could easily understand what figure we are talking about. All figures were commented by the authors.

Reviewer 4 Report

All suggestions have been included. I accept the paper for publication.

Author Response

Comments: All suggestions have been included. I accept the paper for publication.

 Answer: The authors would like to thank the Reviewer for his/her thorough examination of the paper.